# Tectonically-triggered sediment and carbon export to the Hadal zone

Rui Bao[1,2,7], Michael Strasser [1,3,4], Ann P. McNichol[2], Negar Haghipour[1], Cameron McIntyre [1,5,6], Gerold Wefer[4] & Timothy I. Eglinton[1]

Sediments in deep ocean trenches may contain crucial information on past earthquake history and constitute important sites of carbon burial. Here we present [14]C data on bulk organic carbon (OC) and its thermal decomposition fractions produced by ramped pyrolysis/oxidation for a core retrieved from the >7.5 km-deep Japan Trench. High-resolution [14]C measurements, coupled with distinctive thermogram characteristics of OC, reveal hemipelagic sedimentation interrupted by episodic deposition of pre-aged OC in the trench. Low $\delta^{13}C$ values and diverse [14]C ages of thermal fractions imply that the latter material originates from the adjacent margin, and the co-occurrence of pre-aged OC with intervals corresponding to known earthquake events implies tectonically triggered, gravity-flow-driven supply. We show that [14]C ages of thermal fractions can yield valuable chronological constraints on sedimentary sequences. Our findings shed new light on links between tectonically driven sedimentological processes and marine carbon cycling, with implications for carbon dynamics in hadal environments.

[1] Geological Institute, ETH Zurich, 8092 Zurich, Switzerland. [2] National Ocean Science Accelerator Mass Spectrometry Facility, Woods Hole Oceanographic Institute, Woods Hole, MA 02543-1539, USA. [3] Institute of Geology, University of Innsbruck, 6020 Innsbruck, Austria. [4] MARUM-Center for Marine Environmental Sciences University of Bremen, D-28359 Bremen, Germany. [5] Laboratory for Ion Beam Physics, Department of Physics, ETH Zurich, 8093 Zurich, Switzerland. [6] Scottish Universities Environmental Research Centre, Glasgow G75 0QF, UK. [7] Present address: Department of Earth and Planetary Sciences, Harvard University Cambridge, MA 02138, USA. Correspondence and requests for materials should be addressed to R.B. (email: rui.bao@erdw.ethz.ch) or to M.S. (email: Michael.Strasser@uibk.ac.at) or to T.I.E. (email: timothy.eglinton@erdw.ethz.ch)

The deep sea represents one of the most extreme and least-well-studied environments on Earth. Ecological, biogeochemical, and sedimentological processes of the hadal zone (>6 km water depth)[1] are particularly poorly constrained, yet of fundamental importance for understanding the significance of these spatially expansive environments on a global scale[2–6]. Typical hadal environments are deep ocean trenches formed by the downward bending of the oceanic lithosphere at subduction zones along active convergent plate boundary systems, such as around the Pacific rim of fire[7]. Active convergent margins release >90% of the stress accumulated by global plate tectonics in often devastating earthquakes associated with the process of subduction[8]. Recent earthquakes along subduction zones have provided an opportunity to understand dynamical earthquake-triggered sediment remobilization processes in these hadal zones, and to determine spatial-temporal characteristics of such events deposits[9–11]. Key challenges for understanding long-term magnitude-frequency relationships of great subduction zone earthquakes and for assessing their hazard potential are to reliably identify and date earthquake-related event deposits in the geological record, to constrain the provenance and frequency of these events deposits, and to link them to relevant earthquake parameters of the causal seismic events. However, one of the most critical challenges and distinctive feature of the hadal zone environments such as deep-sea trenches, is that the underlying sediments are deposited below the calcite compensation depth (CCD), resulting in an absence of dateable inorganic matter (i.e., carbonate biominerals), thereby confounding traditional radiocarbon ($^{14}C$) dating methods. $^{14}C$ measurements on the associated organic matter (OM), which is not subject to the influence of dissolution of carbonate materials, can provide an alternative approach in these settings. This approach has been previously applied to oceanic environments that are depauperate in carbonate biominerals such as the Antarctic Ocean[12–15]. However, dilution of OM derived from marine productivity with uncertain and variable proportions of pre-aged or petrogenic OM can undermine the validity of this approach.

While there is a growing understanding of sediment supply to the abyssal ocean trenches[5,9,16], our knowledge of the source and composition of associated OM deposited in hadal sediments remains very limited. Only a very small fraction of the OM produced in surface waters escapes remineralization in the water column, settles to the abyssal ocean floor, and is eventually buried[17–19]. In deep ocean trenches associated with subduction zones, OM derived from pelagic sedimentation may be augmented by sediment supply from the adjacent margin. The latter, previously deposited and stored in upslope settings, may be supplied via background sedimentation processes (e.g., bottom and intermediate-depth nepheloid layer transport (BNL and INL)) or via more episodic gravity flows[20]. Thus, OM accumulating in trench sediments may contain a mixture of organic carbon (OC) derived from autochthonous contemporary marine productivity, as well laterally transported OC comprised of continentally derived OC and reworked marine OC. The latter allochthonous inputs may be pre-aged as a consequence of their pre-depositional histories (storage in intermediate reservoirs on the continent and/or the margin), leading to $^{14}C$ age offsets with autochthonous OM. Consequently, hadal zone sediments may contain OM that varies in age and reactivity (bioavailability)[2] as a function of its provenance and mode of supply.

Recently, several new chronological approaches have been developed for marine sedimentary OM that obviate complications from mixed OM sources. For example, compound-specific radiocarbon dating[21,22] has been successfully applied to Antarctic Ocean sediments[14]. However, the application of the method is often limited due to low concentrations of target compounds. Another approach uses a so-called ramped pyrolysis/oxidation (RPO) method in combination with carbon isotopic analysis of the $CO_2$ evolved from thermal decomposition of the sedimentary OM[15,23]. This approach can yield both radiocarbon and stable carbon isotopic information on sedimentary OM components separated according to their thermochemical stability, revealing the spectrum of $^{14}C$ ages, and hence the heterogeneity of OC within a sample. When coupled with new methodologies that allow for high-throughput bulk sediment OC $^{14}C$ determination[24], it may yield more robust constraints on sediment chronologies.

Sediment remobilization induced by the Tohoku-oki earthquake (moment magnitude >9) and associated tsunami that struck NE Japan on 11 March 2011 triggered dense nepheloid layers in the >7 km-deep Japan Trench[9] and resulted in characteristic event deposits in underlying sediments[10,11,25]. The Japan Trench is an oceanic trench formed by the subduction of the oceanic Pacific Plate below the Okhotsk Plate[26]. This plate boundary system is an active seismogenic zone that hosted the most recent tsunamigenic mega-earthquake[27], triggering widespread remobilization and subsequent re-deposition of sediment and associated OM in confined terminal basins with water depths >7 km, far below the CCD[9,10,25].

Building on emerging knowledge of earthquake-triggered sediment remobilization processes gained from this recent earthquake, and the strong temporal constraints on prior similar events[25], the objectives of this study were to assess the role of sediment mobilization and lateral transport processes on the characteristics and $^{14}C$ age of OM in hadal sediments, examine $^{14}C$ age-depth profiles of hadal sediments in the context of episodic, event-driven sedimentation, and assess whether reliable OM-based $^{14}C$ chronologies and age models can be established for carbonate biomineral-poor hadal zone sediments. We explore the utility of the RPO method for determining the radiocarbon ages and carbon isotopic compositions of OM in Japan Trench sediments, and for assessment of remobilization processes along this tectonically active subduction margin. Thermograms and corresponding RPO $^{14}C$ age spectra from sediments sampled at selected depth intervals are presented in the context of a high-resolution bulk OC $^{14}C$ record. The origin and fate of OC, as well as identification of tectonic events that episodically deliver pre-aged OM to Japan Trench sediments are discussed. This constitutes the first application of the RPO method and high-resolution OC $^{14}C$ age profiling to such hadal zone sediments. Our observations reveal translocation and burial of significant quantities of pre-aged OC in the hadal environment, shedding new light on the nature and dynamics of carbon supply to hadal zone.

## Results

**High-resolution bulk OC $^{14}C$ profile.** A high-resolution bulk OC $^{14}C$ depth profile for core GeoB 16431-1 revealed large variations in age, from $1651 \pm 81$ to $9769 \pm 196$ $^{14}C$ yr BP (Figs. 1 and 2, Supplementary Data 1). Marked $^{14}C$ age excursions are evident for two depth intervals (~ 260–425 and ~ 500–616 cm) that coincide with turbidite sequences triggered by the AD 1454 and AD 869 earthquakes, respectively[25,28]. Strong linear fits exist through the three intervening depth intervals defined by Ikehara et al.[25], (Fig. 2; red lines, overall $r^2 > 0.8$).

**In-depth geochemical analysis.** Five samples representing different sedimentation phases (A–E; Fig. 1) were selected for further characterization (Supplementary Table 1). Total OC (TOC) values range from 0.6 to 1.7%, (ave., $1.3 \pm 0.4\%$; $n = 5$), with highest values for sample A and lowest values for samples B and D. Bulk OC $\delta^{13}C$ values range from −30.9‰ to −25.1‰ (ave.

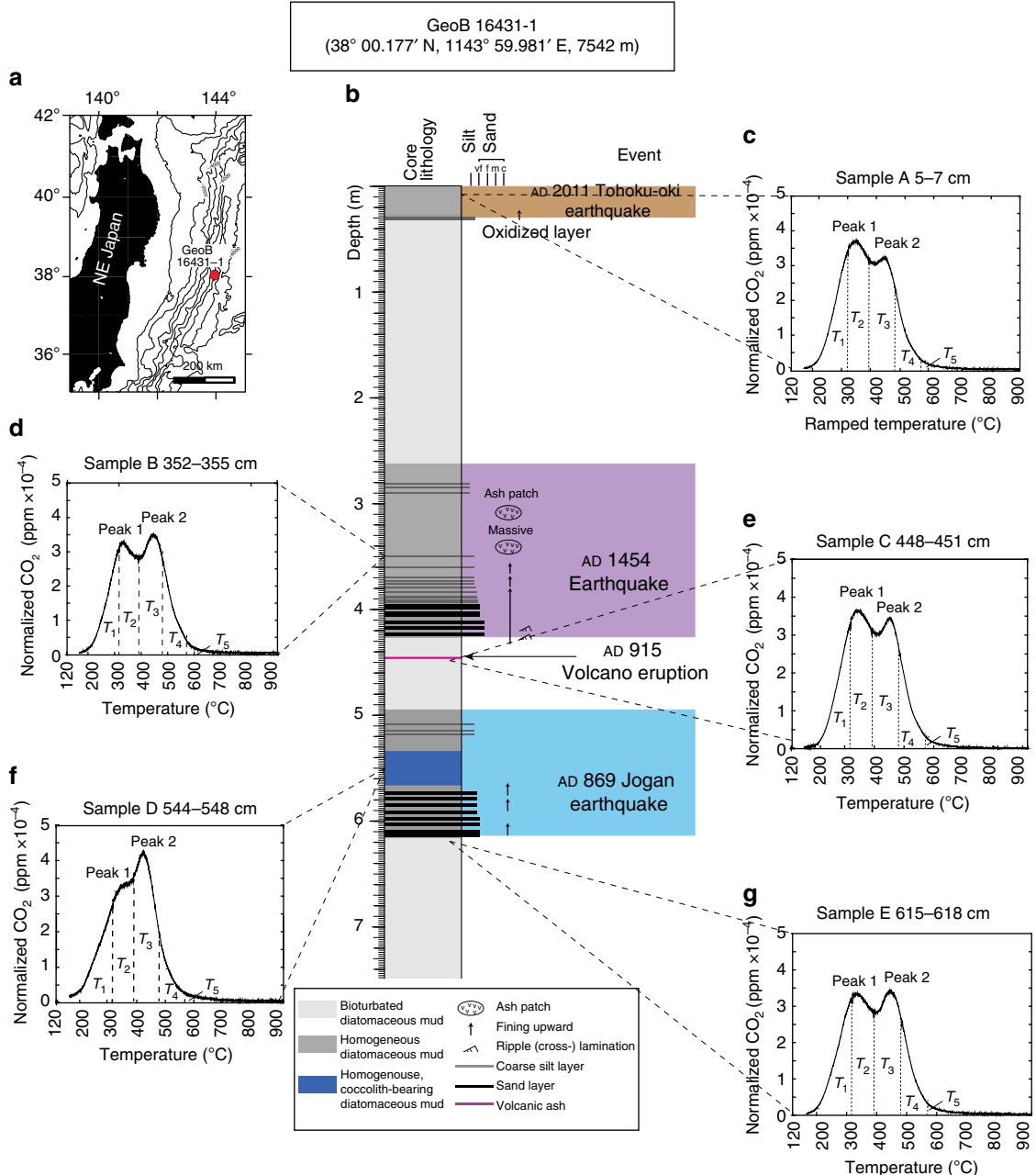

**Fig. 1** Location and lithology of core GeoB 16431-1 in the Japan Trench, and thermograms of RPO of five sub-samples. **a** Map with location of core GeoB 16431-1 (7542 m water depth) in the Japan Trench, where the sediment sequence records historical earthquakes and volcanic eruptions. **b** Lithology of core GeoB 16431-1 and depths of samples (A–E) selected for in-depth study are presented; samples A, B, and D are from inferred earthquake deposits, characterized by graded fine-sand layers fining-upward into homogenous mud, that are sedimentologically distinct from the bioturbated muds that reflect background sedimentation (samples C and E)[46]. The colored boxes show tectonic events[25]. These event deposits correlate with prior earthquakes (AD 2011, AD 1454, and AD 869) and volcanic eruptions (AD 915) documented in Japanese historical records[25]. **c, d, e, f, g** Thermograms from RPO analysis of samples A–E are shown; *Y*-axis is mass-normalized $CO_2$ concentration, *X*-axis indicates temperature gradient

$-27.7 \pm 2.3‰$; $n = 5$), with the lowest values for samples B and D. The $\delta^{13}C$ profile shows relatively consistent values except within the earthquake sequences (Supplementary Fig. 1). Measured bulk OC $^{14}C$ ages (Supplementary Table 1) range from $1850 \pm 107$ yr BP (sample A) to $9425 \pm 228$ yr BP (sample D). The $^{14}C$ age of sedimentary OC from the shallowest core depth (sample A, 5–7 cm, 1850 $^{14}C$ yr BP), corresponding to the period of BNL deposition associated with the 2011 Tohoku-oki earthquake, is markedly older than both that of sinking particulate matter intercepted at 8681 m in the hadal zone of the Japan Trench (~ 250 $^{14}C$ yr BP)[29], and ~ 1900 $^{14}C$ yr older than sediments

immediately below this earthquake sequence. Additionally, the $^{14}C$ age of sample B (352–355 cm; $5218 \pm 146$ yr BP), corresponding to a layer inferred to have been deposited in relation to a major tectonic event[25], is older than sample C (448–451 cm; $3139 \pm 120$ yr BP) deposited earlier. Similarly, the $^{14}C$ age of sample D (544–548 cm; $9425 \pm 228$ yr BP), the third interval hypothesized to reflect a past earthquake event, is older than the underlying sample E (615–618 cm; $3228 \pm 122$ $^{14}C$ yr BP).

The thermograms from RPO analysis of samples A–E in each case show a bimodal distribution with temperatures of maximum $CO_2$ generation ($T_{max}$) of $340 \pm 10$ °C (peak 1) and $450 \pm 8$ °C

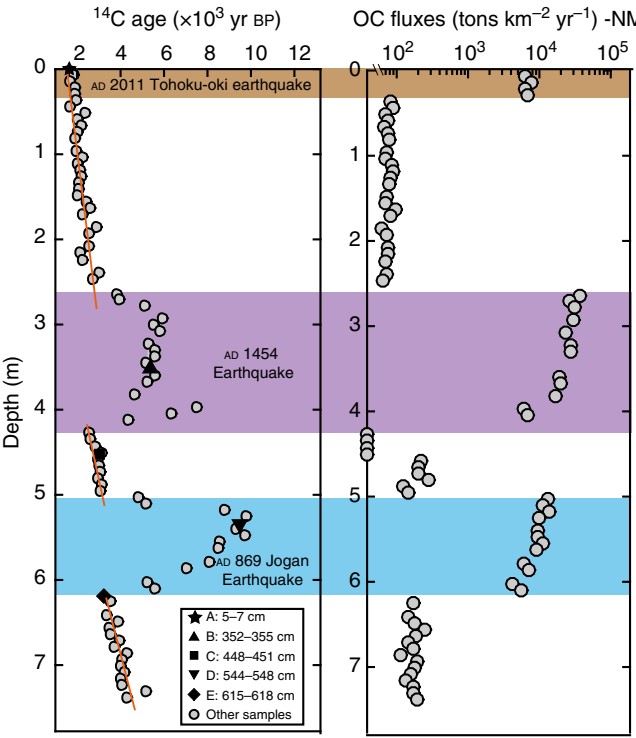

**Fig. 2** Lithology of core GeoB 16431-1. A high-resolution bulk OC $^{14}$C age profile and corresponding OC fluxes (note log scale)

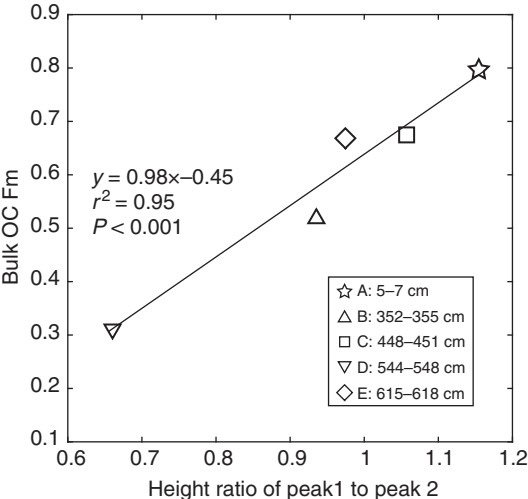

**Fig. 3** Relationship between fraction modern of bulk OC and height ratio of peak 1 to peak 2 in the RPO thermogram

(peak 2), respectively (Fig. 1). While this suggests similarities in overall thermochemical stability, and hence bulk OM characteristics, there are variations in relative peak height indicating differences in proportions of different OM constituents. In samples A, C, and E, the heights of first (lower temperature) peaks are proportionally higher than those of samples B and D (Fig. 3). Stable carbon isotopic values for each thermal fraction were within the range expected for predominantly terrestrial OM (Supplementary Table 1 and Supplementary Fig. 3). In general, the $\delta^{13}$C values decrease from low to high temperatures, ranging from −36.5‰ to −21.9‰ (ave., −26.6 ± 3.7‰, $n$ = 25). The $^{14}$C age of thermal fractions also varies considerably, ranging from 1410 ± 15 to 13553 ± 193 yr BP. In general, lower temperature fractions exhibit younger $^{14}$C ages, with older $^{14}$C ages for higher temperature fractions. Consistently older $^{14}$C ages are evident for sample D, with the highest temperature fraction for this sample ($T_5$) yielding a radiocarbon age of 13553 ± 193 yr BP. In contrast, the $^{14}$C ages of all thermal fractions from sample A (ranging from 1410 ± 15 to 4588 ± 93 yr BP) are younger compared to corresponding fractions from other samples.

## Discussion

Radiocarbon measurements indicate that pre-aged OC is deposited in the Japan Trench, particularly during tectonically triggered sedimentation. Samples B and D have markedly older $^{14}$C ages compared to other intervals (Supplementary Table 1), suggesting entrainment of old OC or enhanced proportions of recalcitrant and old OC due to preferential degradation of labile OC. The first explanation is inconsistent with observations from RPO analysis (Fig. 1), which showed that all five thermogram patterns were similar, suggesting common source(s) of OM to sediments in the Japan Trench. In contrast, the varying proportions of low- and high-temperature peaks are attributed to differing extents of degradation of OM components within each sample. Notably, while the present data set remains limited, a strong linear

relationship is apparent between bulk OC $^{14}$C content (Fm) and the height ratio of peak 1 to peak 2 (Fig. 3), suggesting that bulk $^{14}$C content is largely controlled by the relative proportion of organic components. Such changes in the relative proportion of organic components (peaks) can also be interpreted in the context of selective degradation/preservation of OM. We infer that the relatively old $^{14}$C ages of samples B and D result from longer residence times within diagenetically active reservoirs (water column and surface sediments) and hence greater degradation prior to burial. Stable carbon isotopic characteristics (Supplementary Table 1), specifically the $^{13}$C depletion of bulk OM ($\delta^{13}$C: −25.1‰ to −30.9‰) compared with the typical $\delta^{13}$C values of sinking particulate OM (~−23.0‰ and ~−23.5‰ at 4789 and 8789 m water depth, respectively[29]), suggests that terrestrial OM (or possibly refractory aliphatic macromolecular material[30]) is the dominant source of sedimentary OC in the Japan Trench. In particular, $\delta^{13}$C values of samples B and D (−28.7‰ and −30.9‰, respectively, Supplementary Table 1), corresponding to the AD 1454 and 869 tectonic events, respectively, are ~3‰ lower than those of other samples. Associated OM may therefore be subject to preferential loss of labile, $^{13}$C-enriched marine OM during lateral transport and subsequent sedimentation in the Trench. *Anomalina spp.*, a contemporary shallow-water benthic foraminiferal species, was found in sample B, suggesting sediment transfer and burial following a tectonic event that triggered delivery of continentally derived sediments. This is also consistent with the inferred sedimentation mode[25] whereby earthquake-triggered gravity flows rapidly remobilize and translocate previously deposited sediment and its associated OM from the continental margin to the deep Japan Trench. Gravity flows triggered by tectonic events such as earthquakes (and resultant tsunamis and landslides) lead to relatively high sedimentation rates in these abyssal settings (refs [10,11,25], and references therein). The interseismic hemipelagic deposits that form with high sedimentation rates from 0.8 to >3.0 m kyr$^{-1}$ effectively cover earthquake-induced turbidites and volcanic ash layers and preserve the deposits as a geological record of large tectonic and volcanic events[25]. Ikehara et al.[25] documented thick (~1.5 m), fining-upward turbidite deposits and volcanic ash layers interbedded within bioturbated diatomaceous mud in sediment cores from this region. The terminal basins along the trench floor accommodate the episodic deposition of fine-grained turbidites as graded fine-sand layers fining-upward into homogenous diatomateous mud. In general, coarser-grained sediments sink more

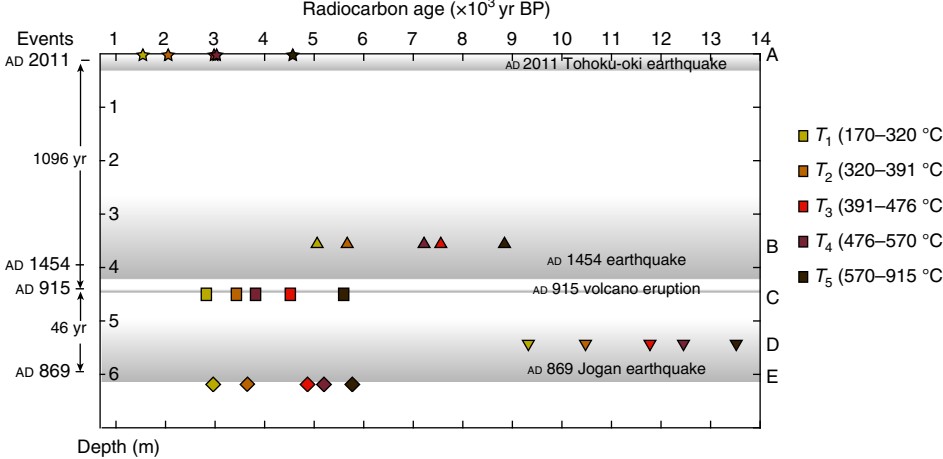

**Fig. 4** Chronology and radiocarbon characteristics of RPO thermal fractions of OM in sediment core GeoB 16431-1 for samples A–E. The samples' labels are shown adjacent to right-hand Y-axis and in Fig. 3. Adjacent to the right-hand Y-axis are the thermal decomposition temperature intervals. The gray shading denotes the earthquake events. Their names and extents are marked by arrows on the left-hand Y-axis. The X-axis (top) indicates the $^{14}$C age (yr BP) of thermal decomposition fractions (the measurement error is smaller than the symbol)

rapidly, whereas the finer-grained sediments settle more slowly, resulting in upward-fining of sediments within the event deposit (Fig. 1, lithology). Indeed, some fractions of the sediments introduced into suspension may reside for protracted periods of time within the BNL following an episodic event[31,32]. Samples B and D are from the upper (finer-grained) zone, implying later-stage deposition (Fig. 1), and protracted entrainment within BNLs (or INLs).

The relatively invariant temperatures for maxima of the first and second RPO peaks ($T_{max1}$ and $T_{max2}$; $340 \pm 10$ and $450 \pm 8$ °C, respectively) suggest that there is little difference in thermostability, and by inference the characteristics of the OM, due to post-depositional diagenetic processes in the Japan Trench sediment core. The nature of OC (e.g., lability vs. recalcitrance) can be indirectly inferred based on thermal stability[33]. The samples investigated here have peaks presumably corresponding to the "labile" and "recalcitrant" OC categories. The relatively old $^{14}$C ages of corresponding thermal fractions $T_2$ (320–391 °C) and $T_3$ (391–476 °C) (ranging from $2050 \pm 85$ to $12164 \pm 145$ yr BP) suggest burial of old yet both relatively labile and recalcitrant OM in the Japan Trench. Preferential degradation of young, labile OM may lead to an increase bulk $^{14}$C ages. Figure 4 shows the $^{14}$C age spectrum of thermal fractions among the five samples investigated. The $^{14}$C age difference ($^{14}$C heterogeneity) between the most labile (lowest temperature, $T_1$) and the most refractory (highest temperature, $T_5$) fraction exceeds 4000 yr in samples B and D. In contrast, samples C and E (inferred background sedimentation, deposited around AD 915 and 869, respectively) exhibit smaller and more consistent age offsets ($2714 \pm 118$ and $2820 \pm 122$ yr, respectively; Fig. 4). Preferential degradation also enhances $^{14}$C age heterogeneity (i.e., samples B and D). The degree of $^{14}$C age heterogeneity may depend on the pre-depositional history and/or extent of in situ microbial degradation of younger OC.

The high-resolution bulk OC $^{14}$C profile provides further insights into temporal variability in sedimentary OM accumulation in the Japan Trench (Fig. 2). Two depth intervals that are characterized by markedly older $^{14}$C ages, ~ 260–425 and ~ 500–616 cm, correspond well to sedimentological layers attributed to the AD 1454 and AD 869 earthquakes, respectively[25], implying enhanced (re)burial of OC in the trench associated with these events. This is also consistent with RPO results that indicates greater accumulation of old OC in the Japan Trench

triggered by tectonic events. These older ages could be a consequence of protracted storage in intermediate reservoirs on land (e.g., soils) and/or on the continental margin. For example, OM in soils and exported to the ocean from rivers can exhibit apparent residence times >1000 yr[34,35]. OM can reside within continental margin surficial sediments for millennia[36]. The sediment drape over continental slopes can be destabilized by tectonic activity[11] and mobilized sediments may remain entrained in suspension-deposition cycles and subject to widespread dispersal via nepheloid layer transport for several hundred to several thousand years[36–38] prior to eventual re-deposition. These processes may be accompanied by OM degradation and $^{14}$C aging. Irrespective of the specific process(es) at play, these results indicate that remobilization of earthquake-triggered sediments results in enhanced burial of pre-aged OC in the Japan Trench (Fig. 2, OC fluxes).

The bulk $^{14}$C age of the uppermost sediment layers from the gravity core (ave. 1811 yr BP, $n = 3$, top 20 cm) is ~ 2000 yr older than the date of sediment emplacement following the AD 2011 Tohoku-oki earthquake. The markedly older $^{14}$C ages of OC in sediments deposited coincident with prior tectonic events also implies significant and time-varying inputs of margin-derived pre-aged OC (Fig. 2). These variable contributions of allochthonous OC confound to develop $^{14}$C-based chronologies for sedimentary sequences deposited in the hadal zone, and preclude direct age assignments for specific event layers. The influence of aged OC in Japan Trench sediments is not restricted to event-related deposition (Fig. 2). For instance, the documented tephra age is AD 915, whereas the bulk OC $^{14}$C age just below the tephra layer is $3139 \pm 120$ yr BP, clearly exceeding the estimated local marine reservoir age of 830 yr[39]. Background sedimentation (i.e., excluding the event layers) is characterized by quasi-linear relationships between bulk OC $^{14}$C age vs. sediment depth (Fig. 2, red lines), suggesting that in the absence of major tectonic events, similar sedimentological and diagenetic conditions prevail over the duration of the record, and implying that OM originates from common or similar sources. However, development of sediment chronologies for Japan Trench sediments based on bulk OC $^{14}$C age is prone to uncertainty due to variable and uncertain proportions of aged OC.

Although bulk OC $^{14}$C ages of Japan Trench sediments are insufficiently constrained to derive chronologies, it may be possible to use $^{14}$C ages of specific (thermal) fractions from RPO of

samples derived from different depth horizons to establish a relative chronology. The RPO results reveal that the lowest temperature fractions ($T_1$, labile fraction) are older than the known [14]C ages of the tectonic events. For example, the conventional [14]C age of $T_1$ from sample C (below the AD 915 volcano eruption) is 2830 ± 20 yr BP. While it is not presently possible to accurately determine the age and proportion of autochthonous and allochthonous OC in a sample, precluding use of conventional [14]C ages of both bulk OC and $T_1$ fraction as a absolute dating tools, [14]C age differences between specific thermal fractions may provide an approximation of age differences between sediment intervals, particularly for sediments deposited within the last ~ 3000 yrs, during which radiocarbon ages closely parallel calendar ages (INTCAL13[40]). We note that the [14]C age difference (120 ± 45 yr) between $T_1$ fractions in samples C and E approaches the time offset (46 yr) between the volcanic eruption in AD 915 and Jogan earthquake in AD 869 (Fig. 4), particularly considering sampling uncertainty (samples C and E lie below each event sequence). Similarly, the age offset (126 ± 195 [14]C yr) between $T_5$ fractions from these two samples approximates the known time offset between the two tectonic events. The consistency in [14]C differences (in Fig. 4, if connected, they would be "parallel" lines) between sedimentary layers of samples C and E for both the youngest thermal fraction ($T_1$) and oldest thermal fraction ($T_5$), and their correspondence with known tectonic and volcanic events, implies that these thermal fractions are not affected by preferential OM degradation during subsequent burial, while the lack of observed consistency in [14]C ages of $T_3$ fractions is likely because this temperature fraction is sensitive to subtle variations in proportions of different organic components that contribute to this specific temperature interval (391–476 °C). Overall, [14]C age differences between corresponding thermal fractions from different sediment intervals are inferred to primarily reflect radioactive decay following sedimentation. Considering measurement errors and additional uncertainties, we believe this approach holds promise for placing chronological constraints on Japan Trench sediment cores. Age models could be constructed provided these [14]C ages can be anchored to at least one sample of known calendar age (e.g., dated tephra layer).

It is interesting to note that the [14]C age of OC in the uppermost sediment interval investigated, which represents deposition following the recent Tohoku earthquake (AD 2011), is relatively young compared to the other layers attributed to previous earthquakes (AD 1454 and 869). This could be explained if finegrained sediments carrying older OM remain in suspension for a long time prior to eventual deposition[9,32]. Alternatively, variations in provenance of sediments delivered to the Japan Trench, which depends on the upslope location(s) of sediment failure and gravity flow, may be responsible for these different age characteristics. Finally, old OC ages could be masked by supply of recent OM that has inherited an elevated bomb radiocarbon signal via photosynthetic carbon fixation in the surface ocean[41]. Further investigation of sedimentary processes following the most recent earthquake is warranted in order to be able to recognize, and to constrain the frequency and origin of past tectonic events in hadal sedimentary sequences.

This first application of RPO and [14]C analysis of specific thermal decomposition fractions provides new chronological constraints on past depositional events in the Japan Trench. The approach is particularly useful when coupled with high-resolution bulk OC [14]C measurements that document when background hemipelagic sedimentation is interrupted by tectonically triggered gravity flows. Our observations highlight the large-scale translocation and burial of terrigenous materials in the hadal zone associated with these events. This lateral carbon pump has implications for nature and dynamics of carbon supply to abyssal ocean subduction zone sediments, and for associated benthic communities[2]. More broadly, this approach holds promise for development of chronologies for hadal and other sediments that lack microfossils for conventional radiocarbon dating and isotope stratigraphy.

## Methods

**Sampling strategy and preparation**. We analyzed gravity core GeoB 16431-1 (9.4 m recovery), which was retrieved from a terminal Japan Trench basin during R/V SONNE cruise SO219A (ref. [28]; sampling site 38° 00.177′ N, 143° 59.981′ E; 7542 m water depth; Fig. 1, map). The core was split, described, and sectioned on board. The sediment core sections were then shipped and stored at 4 °C in the core repository of MARUM-Center for Marine Environmental Sciences at the University of Bremen, and subsequently subsampled and stored at −20 °C at ETH Zurich until further processing.

A suite of 82 sediment samples (representing sampling interval of ~ 7.5 cm; Supplementary Data 1) was selected for construction of a high-resolution profiles of bulk OC content and [14]C age. Samples were freeze-dried in pre-combusted vials, and aliquots were weighed into Ag capsules for fumigation with concentrated HCl (37% Trace-Metal purity, 60 °C, 72 h) to remove inorganic carbon. The acidified samples were subsequently neutralized with NaOH under the same conditions (72 h). Sample preparation was performed in the Biogeoscience Group Laboratories at ETH Zurich (Fig. 2; Supplementary Data 1).

Five sub-samples from specific depth intervals were selected for in-depth analysis (Supplementary Table 1). The sampling strategy was guided by the interpreted event stratigraphy of this core[25], inferred sediment remobilization events (i.e., sample A (5–7 cm) deposited following the 2011 Tohoku earthquake; samples B (352–355 cm) and D (544–548 cm) from older event deposits linked to historical earthquakes[25]); and of intervals of quasi-continuous background sedimentation in the vicinity of marker beds with known depositional age (sample C (448–451 cm) just above the To-a volcanic ash layer of AD 915, and sample E (615–618 cm) just below the event layer correlated to the Jogan Earthquake of AD 869[25]; Fig. 1). Samples for RPO measurements were prepared in the same manner as above, but in the National Ocean Science Accelerator Mass Spectrometry facility (NOSAMS) at Woods Hole Oceanographic Institution.

**Bulk organic geochemical properties**. TOC content of the five selected sediment samples was determined using established protocols at NOSAMS[42]. Freeze-dried samples (~ 3 g dry weight) were weighed into pre-combusted glass Petri dishes (5 cm diameter). A beaker filled with ~ 30 mL HCl (37%, Trace-Metal purity) was placed at the bottom of a 250 mm inner diameter glass desiccator; the samples were placed on a ceramic tray above the acid beaker. The desiccator was evacuated and samples were treated at 60 °C for 72 h. After fumigation, excess acid was neutralized through replacing the acid beaker with ~ 20 g NaOH pellets in a precombusted Petri dish. The desiccator containing the acidified samples with NaOH pellets was again evacuated and placed in an oven (60 °C, 72 h). Approximately 7% of the purified $CO_2$ gas was split on a vacuum line and used for stable isotope determination via isotope ratio mass spectrometry (IRMS), with the remainder graphitized using standard methods at NOSAMS[43]. Corresponding stable carbon isotopic ($\delta^{13}C$) values were determined to a precision of better than ±0.1‰, and are reported relative to Peedee Belemnite. Three samples listed in Supplementary Table 1 were measured for [14]C at NOSAMS, while the other two were analyzed for [14]C using a MICADAS accelerator mass spectrometer (AMS) system at ETH Zurich and calibrated against standard Oxalic Acid II (NIST SRM 4990C) and inhouse radiocarbon blank $CO_2$ or anthracite coal. Samples for [14]C analysis at ETH Zurich were measured directly as $CO_2$ gas. [14]C precision for gas was better than ±10‰ on a modern standard. The high-resolution [14]C ages of 82 bulk samples' analysis and 5 bulk samples from specific depth intervals were measured using a coupled elemental analyzer (EA)/IRMS/AMS online system at ETH Zurich[24]. Radiocarbon data are reported as fraction modern (Fm) and radiocarbon age[44].

**RPO measurements**. Ramped temperature pyrolysis/oxidation was performed on the five bulk sediment samples selected for in-depth analysis (Supplementary Table 1; Fig. 4). As in previous work[15,23], acidified samples (~ 100–200 mg) were loaded into a quartz reactor and subjected to a constant rate of temperature increase (5 °C min$^{-1}$) until 915 °C. Evolved components were simultaneously oxidized and the resulting gases (e.g., $CO_2$, $SO_2$) were purified by passage through a chemical reactor under isothermal conditions[45]. $CO_2$ concentrations and the resulting thermograms were obtained using a flow-through infrared $CO_2$ analyzer (Sable Systems International Inc., CA-10a). Evolved thermal components were integrated over five temperature intervals $T_n$ ($T_1$: 170–320 °C; $T_2$: 320–391 °C; $T_3$: 391–486 °C; $T_4$: 486–570 °C; and $T_5$: 570–915 °C; Fig. 1c–g). A leak check was conducted at the beginning and every hour during the experiment. $CO_2$ samples corresponding to individual thermal fractions were sealed in pre-combusted glass tubes with copper oxide and silver balls for combustion to purify gases prior to [14]C measurement.

**OC fluxes**. OC fluxes were calculated using measurements of sample density, TOC content, and sedimentation rate (OC flux = TOC×density×sedimentation rate). Sample density (bulk density, $g\,cm^{-3}$) was measured by gamma-ray attenuation density analyses from multi-sensor-core logging at MARUM (Supplementary Data 1). TOC was measured using the EA/IRMS/AMS system at ETH Zurich (data shown in Supplementary Data 1). Sedimentation rate was calculated based on the ratio of sediment depth spanning time intervals constrained by known events. Sedimentation rate for the 2011-earthquake turbidite sequence was calculated by the ratio of depth of sequence (~ 30 cm) to 1 yr (time interval between 2011 and the collection year, ~ 30 cm yr$^{-1}$). Sedimentation rates for the other two earthquake turbidite sequences were estimated by multiplying ~ 30 cm yr$^{-1}$ and each sequence depth (assuming that every earthquake sequence exhibits rapid sedimentation during 1 yr). Due to compaction during the sedimentation, our approach may yield underestimates of the OC fluxes associated with the AD 1454 and AD 869 earthquakes.

**Chronological considerations**. In the section of this paper addressing the development of Japan Trench sediment chronologies, we focus on $^{14}$C offsets between thermal fractions (e.g., samples C and E) in order to derive a relative, rather than absolute chronology, and highlight its potential relevance to other hadal zone sedimentary sequences lying below the CCD that lack microfossils for conventional radiocarbon dating and isotope stratigraphy. Although the differences in isotopic compositions between bulk OC and RPO data are significant, they are roughly constant (small, Fm values ~ 0.06–0.08, ±0.02; Supplementary Fig. 2). As such, the isotopic discrepancy does not impact the chronological aspect of the paper and specifically the validity of chronologies developed based on RPO. Considering the mass contribution of target thermal fractions to each sample and the associated $^{14}$C measurement errors (ave. ±~100 $^{14}$C yr, based on ETH measurement), the offset (relative value) between $^{14}$C ages of thermal fractions is relatively small (±~200 $^{14}$C yr), and does not undermine our statement in the manuscript. In the manuscript, we acknowledge this time difference (i.e., relative offset: 120 ± 45 yr, time offset: 46 yr). We emphasize that the novelty of our approach is to first identify which samples (background sedimentation) are appropriate for further chronological analysis through bulk OC $^{14}$C measurement, and use information derived from RPO to constrain $^{14}$C age offsets (of $T_1$ fractions) between sediment layers. This yields relative ages that can be anchored based on one or more known events (calendar age). Furthermore, while note that the presented $^{14}$C ages are not calendar ages (documented event time), during the late Holocene (which constitutes the time interval of primary interest in this study), measured $^{14}$C ages closely parallel calendar ages[40]. Thus, accuracy of the relative chronology is dependent on measurement precision and offsets between calendar and $^{14}$C ages, we consider this constitutes a unique approach for documenting the timing and frequency of event deposits pre-dating historical records in the hadal zone.

**Data availability**. Data sets generated during and/or analyzed during the current study can be found in Supplementary Table 1 and Supplementary Data 1, and are also available from the corresponding author on reasonable request.

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

## Acknowledgements

We thank the captain and crew of R/V SONNE for onboard assistance during cruise SO219A in 2012. This cruise is supported by Federal Ministry for Education and Research, and Deutsche Forschungsgemeinschaft. This study is supported by Doc. Mobility Fellowship (P1EZP2_159064) (R.B.) from the Swiss National Science Foundation (SNSF). This work is also supported by SNF "CAPS-LOCK" project 200021_140850 (T.I.E.), by SNSF grant (133481) (M.S.), and Austrian Science Foundation (P 29678-N28) (M.S.). We thank Dr Lukas Wacker for helpful comments. We thank support of the NOSAMS staff in the execution of this project. We also thank Ms. Chen in Tongji University for assistance in identification of foraminifera. We greatly appreciate the assistance from members of the Laboratory for Ion Beam Physics in all aspects of the AMS measurements.

## Author contributions

T.I.E., M.S. and R.B. conceived and designed the project. R.B. wrote the paper with input from all co-authors and produced the figures. M.S. and G.W. collected the samples. R.B., T.I.E. and M.S. interpreted results. R.B., N.H., and C.M. carried out carbon isotope analyses. R.B. and A.P.M. carried out the RPO analyses.

## Additional information

**Competing interests:** The authors declare no competing financial interests.

