## [Peer Review File · Nature Communications]

Reviewers' comments:

Reviewer #1 (Remarks to the Author):

The history of paleoearthquakes is critical for better evaluation of geohazards in tectonically active margins. The evolving field of submarine paleoseismology seeks to document historic and prehistoric earthquake ruptures from the sedimentation record. But the great depths of trenches at subduction systems are a major problem due to a lack of carbonate biominerals for dating events. Developing a chronology by dating of the organic matter in the sediment is problematic because results commonly show mixing of marine and continental sources, remineralization and dilution resulting from organic matter derived from marine productivity. New methods are being developed by the scientific community to date and characterize the sources of resedimented deposits in order to better understand the signature of catastrophic earthquake and tsunami such as the 2011 Tohoku-oki. The Bao et al paper addresses these issues with innovative techniques that have the potential of transforming studies of transported sediments not just in tectonically active margins below the Carbonate Compensation Depth (CCD) but also in passive margins and in other settings where calcium carbonate minerals are lacking.

Bao et al's technique is novel and backed up by solid data. Their results correlate with previously published ages based on tephra chronology and tectonic sedimentation events derived from the lithology (Ikehara et al., 2016). Bao et al use two methods to address dating and provenance of organic matter. Ramped Pyrolysis Oxidation (RPO) in combination with carbon isotopic analyses and thermal decomposition of the organic matter (OM). These results are coupled with high-resolution bulk radiocarbon ages. They also measured the sediment total organic content, sediment density and calculated sedimentation rates and organic carbon fluxes. Intervals of sediment reworking in their studied core (GeoB16431-1 recovered at 7.5 km of water depth) reveal a variety of ages in two distinct intervals. These intervals contain mixed organic carbon (OC) derived ages and correlate with previously dated event deposits linked with two historic earthquakes and tsunami that occurred in AD 1454 and AD 869 (Ikehara et al., 2016).

The RPO measurements applied to five samples documented: 1) a bimodal distribution with the lower temperature peaks proportionally higher than higher temperature peaks, 2) that the lower temperatures are associated with younger ages, and 3) $\delta^{13}\text{C}$ values decrease (more depleted) with increased temperatures. The lower $\delta^{13}\text{C}$ values measured in the intervals with event deposits, when compared to marine, suggest that terrestrial organic matter is a dominant source. Indicating sediment remobilization during earthquakes and providing a marker within which to better characterize intervals of transported sediment. The RPO method also shows that the temperature doesn't vary a lot indicating a common organic matter source.

Bao et al findings are groundbreaking because they demonstrate that thermal differences can provide age approximations between sedimentation events, and that once buried there is no preferential degradation of organic matter. Both are very important. The fact that the ages may reflect the radiocarbon radioactive decay and the age of deposition is also critically important for subduction margins below the CCD and other settings that lack inorganic carbon for dating. The dating of organic matter in sediments has been challenging but the results as presented in this study are very convincing and have the potential of generating new fields or research. This study is important from the sedimentation point of view because it allows studying the stratigraphic record in terms of event deposits triggered by earthquakes and tsunami by linking radiocarbon ages to sediment remobilization and showing a strong correlation with turbidite event deposits documented from the lithology. For all these reasons, I strongly support the publication of Bao et al manuscript in Nature Geoscience.

Minor edits to the text and figure captions.

Line 257 please delete "have"

Line 269 change "that" to than

Line 273 remove "confounding" enter "confounds"

The figure 4 caption, needs additional information. I suggest modifying.

Line 450 ...for samples A-E, the labels are shown on Figure 3. Adjacent to the right-hand Y axis are

the thermal decomposition temperatures.

Line 451....the gray shading notes the tectonic event. Its name and extent are marked by arrows on the left Y axis

Reviewer #2 (Remarks to the Author):

Bao et al., present new high-resolution ^{14}C data from sediments deposited in the Japan Trench over approximately the last millennia. Using bulk and thermal decomposition radiocarbon analysis they suggest that earthquakes trigger the delivery and burial of pre-aged terrigenous organic carbon in the Hadal zone. This is a very nice study that tackles an important scientific question. I am impressed by the density of radiocarbon data (figure 2, middle panel is amazing!) and I find the thermal decomposition analysis novel and powerful. That said, I believe that the overall layout of the manuscript is not optimal. Specifically I suggest focusing the manuscript on its main findings and developing the discussion around the consequences of the author's findings with respect to the carbon cycle. I am not convinced by the current discussion regarding chronology and I suggest dropping this part of the manuscript. I also have a host of more specific comments and questions that will need to be addressed before the manuscript can be accepted.

Specific comments (in order of appearance)

L23: Hadal may be too technical for the abstract in a multidisciplinary journal.

L66: define "fossil" or use a better term (e.g. petrogenic).

L124: reference format to be fixed.

L129: tell the reader how long the core is.

L139: given that these are downcore data I don't get why you're reporting an average age. Also, make it clear that these are ^{14}C ages (I assume they are).

L143: how did you define the boundaries of the 3 intervals? Based on a statistical test? Which one?

L148-149: why are you reporting TOC and $\delta^{13}\text{C}$ for these 5 samples only? It would be very useful to report and discuss $\delta^{13}\text{C}$ for the entire data set.

L150: given analytical uncertainty on $\delta^{13}\text{C}$ values, round all values to 1 decimal

L151-152: this is where it would really help to report $\delta^{13}\text{C}$ for the entire data set. How do the 5 samples compare with the rest of the data, especially with the hemipelagic intervals?

L154-158: why not comparing with the sediment immediately below the 2011 earthquake deposits? That would be much more straightforward!

Thermal decomposition radiocarbon analysis: how does the weighted average composition (^{13}C and ^{14}C) of the fractions compare with the bulk measurements?

L230-234: these divisions are arbitrary and in my opinion not supported by the data presented in the paper the authors are referring to.

L243: looking at figure 2, samples C and E do not appear to fall on the “linear trends” characteristic of the hemipelagic sedimentation, they have systematically higher ^{14}C ages. How can they be considered “background” then?? This is a clear weakness in my opinion, as it stands it doesn’t look like the authors have actually analyzed the thermal reactivity (and associated ^{14}C composition) of any truly “background” sediment.

L377-379: please provide details on this discrepancy, is this why samples C and E look like they have higher ^{14}C ages than the background?

OC fluxes: provide details regarding these estimates (specifically sedimentation rates). In general I am not a fan of calculating fluxes based on a 1D description of sediment accumulation.

Figure 3: The y axis needs to be normalized to account for aging (i.e. sample E was deposited more than 1000 years before sample A, obviously affecting the bulk ^{14}C difference between these samples). The x axis isn’t optimal either. Peak height is never the best choice when looking at complex “chromatograms” because of non-linear effects. Areas would be much better.

Figure 4: Here you should also normalize the x axis to account for the aging effect (see comment above).

Reviewer #3 (Remarks to the Author):

This is an interesting paper that presents a data set on the bulk radiocarbon (^{14}C) age of organic carbon in sediments within the Japan Trench along with the ^{14}C ages and stable carbon isotopic compositions of thermal decomposition fractions produced by Ramped Pyrolysis/Oxidation. The paper is novel both in its presentation of detailed information on the ages of earthquake-induced event deposits in such a deep marine (hadal) setting and especially because of the application of the cutting-edge RPO technique. The paper could potentially be of interest to a wide spectrum of researchers, including those with primary interests in paleoseismology, sedimentary geochronology, and marine carbon cycling. Notably, however, the abstract states that the findings will “shed new light on the nature, frequency, and magnitude of past tectonic events,” yet I see no obvious indication of any contributions to those solving questions.

In my view, some of the conclusions of this paper are well supported, but others are in need of further explanation. The authors show that the bulk ^{14}C ages of three event deposits, which record earthquakes that occurred in 869, 1454, and 2011 AD, are thousands of years older than their ages of deposition. Based on RPO analyses of 5 samples, they conclude that the bulk ^{14}C ages must reflect the mobilization during these earthquakes of terrestrially-derived organic carbon that was “pre-aged” in the water column or seabed. These are interesting findings and I have no quibble with these conclusions. What is less clear to me (and in fact is counter-intuitive), is how pre-aging and consequent loss of young labile OM would increase the heterogeneity of ^{14}C ages in earthquake deposits. A clearer explanation is required. Furthermore, in my view the idea that comparison of the ages of specific thermal fractions at various levels in stratigraphic sequences could be used to develop age models for deep sea sediments is interesting, but highly speculative given that it is based on results from only two samples. The fact that this doesn’t appear to work at all with one of the thermal fractions (T3) is problematic and unexplained. Finally the suggestion that bomb carbon may influence the age of even the most refractory thermal fraction in the most recent (Tohoku) earthquake layer in the Japanese Trench is contrary to the prior conclusion that this fraction is highly aged and requires clarification as well.

The figures are good but would benefit from some improvements. The gray tones used to show background sediments and event layers in Figures 1 and 2 are too similar and are hard to see, and the blue bar is unexplained. Volcanic is misspelled in Figure 2.

Response to Reviewers' comments

In order to further clarify our Responses, all responses that are not quotes from the main manuscript in the new version are written in *Italics*.

Reviewer #1: The history of paleoearthquakes is critical for better evaluation of geohazards in tectonically active margins. The evolving field of submarine paleoseismology seeks to document historic and prehistoric earthquake ruptures from the sedimentation record. But the great depths of trenches at subduction systems are a major problem due to a lack of carbonate biominerals for dating events. Developing a chronology by dating of the organic matter in the sediment is problematic because results commonly show mixing of marine and continental sources, remineralization and dilution resulting from organic matter derived from marine productivity. New methods are being developed by the scientific community to date and characterize the sources of resedimented deposits in order to better understand the signature of catastrophic earthquake and tsunami such as the 2011 Tohoku-oki. The Bao et al paper addresses these issues with innovative techniques that have the potential of transforming studies of transported sediments not just in tectonically active margins below the Carbonate Compensation Depth (CCD) but also in passive margins and in other settings where calcium carbonate minerals are lacking.

Bao et al's technique is novel and backed up by solid data. Their results correlate with previously published ages based on tephra chronology and tectonic sedimentation events derived from the lithology (Ikehara et al., 2016). Bao et al use two methods to address dating and provenance of organic matter. Ramped Pyrolysis Oxidation (RPO) in combination with carbon isotopic analyses and thermal decomposition of the organic matter (OM). These results are coupled with high-resolution bulk radiocarbon ages. They also measured the sediment total organic content, sediment density and calculated sedimentation rates and organic carbon fluxes. Intervals of sediment reworking in their studied core (GeoB16431-1 recovered at 7.5 km of water depth) reveal a variety of ages in two distinct intervals. These intervals contain mixed organic carbon (OC) derived ages and correlate with previously dated event deposits linked with two historic earthquakes and tsunami that occurred in AD 1454 and AD 869 (Ikehara et al., 2016). The RPO measurements applied to five samples documented: 1) a bimodal distribution with the lower temperature peaks proportionally higher than higher temperature peaks, 2) that the lower temperatures are associated with younger ages, and 3) $\delta^{13}\text{C}$ values decrease (more depleted) with increased temperatures. The lower $\delta^{13}\text{C}$ values measured in the intervals with event deposits, when compared to marine, suggest that terrestrial organic matter is a dominant source. Indicating sediment remobilization during earthquakes and providing a marker within which to better characterize intervals of transported sediment. The RPO method also shows that the temperature doesn't vary a lot indicating a common organic matter source.

Bao et al findings are groundbreaking because they demonstrate that thermal differences can provide age approximations between sedimentation events, and that once buried there is no preferential degradation of organic matter. Both are very important. The fact that the ages may reflect the radiocarbon radioactive decay and the age of deposition is also critically important for subduction margins below the CCD and other settings that lack inorganic carbon for dating. The dating of organic matter in sediments has been challenging but the results as presented in this study are very convincing and have the potential of generating new fields or research. This study is important from the sedimentation point of view because it allows studying the stratigraphic record in terms of event deposits triggered by earthquakes and tsunami by linking radiocarbon ages to sediment remobilization and showing a strong correlation with turbidite event deposits documented from the lithology. For all these reasons, I strongly support the publication of Bao et al manuscript in Nature Geoscience.

Response: *We appreciate these positive comments and the strong recommendation to publish our manuscript in a Nature system journal.*

Minor edits to the text and figure captions.

Line 257 please delete "have"

Response: *We have deleted it; see line 259 of new version.*

Line 269 change “that” to than

Response: *We have changed it; see line 271 of new version.*

Line 273 remove “confounding” enter “confounds”

Response: *We have revised it; see line 275 of new version.*

The figure 4 caption, needs additional information. I suggest modifying. Line 450 ...for samples A-E, the labels are shown on Figure 3. Adjacent to the right-hand Y axis are the thermal decomposition temperatures. Line 451...the gray shading notes the tectonic event. Its name and extent are marked by arrows on the left Y axis

Response: *In the caption for figure 4 of the new version, we have re-written it as follows: “Chronology and radiocarbon characteristics of RPO thermal fractions of OM in sediment core GeoB 16431-1 for samples A – E (labels shown adjacent to right-hand Y axis and in Figure 3). Adjacent to the right-hand Y axis are the thermal decomposition temperature intervals. The gray shading denotes the earthquake events. Their names and extents are marked by arrows on the left-hand Y axis. The X axis (top) indicates the ¹⁴C age (yr BP) of thermal decomposition fractions (the measurement error is smaller than the symbol).”*

Reviewer #2: Bao et al., present new high-resolution 14C data from sediments deposited in the Japan Trench over approximately the last millennia. Using bulk and thermal decomposition radiocarbon analysis they suggest that earthquakes trigger the delivery and burial of pre-aged terrigenous organic carbon in the Hadal zone. This is a very nice study that tackles an important scientific question. I am impressed by the density of radiocarbon data (figure 2, middle panel is amazing!) and I find the thermal decomposition analysis novel and powerful.

Response: *We appreciate these positive comments on the novelty and scientific significance of our study.*

That said, I believe that the overall layout of the manuscript is not optimal. Specifically I suggest focusing the manuscript on its main findings and developing the discussion around the consequences of the author’s findings with respect to the carbon cycle. I am not convinced by the current discussion regarding chronology and I suggest dropping this part of the manuscript. I also have a host of more specific comments and questions that will need to be addressed before the manuscript can be accepted.

Response: *We agree that the carbon cycle implications are certainly very important and interesting. Indeed, this is a line of research that we are pursuing as part of follow-up investigations. Nevertheless, it would require a major reorganization to shift the focus in this new manuscript from the current emphasis on the advances in developing chronologies for Hadal zone sedimentary sequences in the context of documenting past tectonically-triggered deposition. Additionally, we note that the other reviewers are enthusiastic about the paper with its present focus, and comment that it is “of interest to a wide spectrum of researchers”. Thus, we prefer to keep the present format but with some revisions in the new manuscript. In the following, we response the comments and concerns on a point-by-point basis.*

Specific comments (in order of appearance)

L23: Hadal may be too technical for the abstract in a multidisciplinary journal.

Response: *We changed the “Hadal sediment” as “trench sediment”; see line 24.*

L66: define “fossil” or use a better term (e.g. petrogenic).

Response: *We now use “petrogenic”; see line 67.*

L124: reference format to be fixed.

Response: *We corrected it; see line 126.*

L129: tell the reader how long the core is.

Response: *We add the “9.4 m recovery”; see line 131.*

L139: given that these are downcore data I don’t get why you’re reporting an average age. Also, make it clear that these are 14C ages (I assume they are).

Response: *We deleted the expression “ave. 4007 ±105 yr BP, n = 82,”.*

L143: how did you define the boundaries of the 3 intervals? Based on a statistical test? Which one?

Response: *We defined them based on the classification of Ikehara et al. (2016). In line 145, we revised the text as follows: “Linear fits through the three intervening depth intervals from Ikehara et al. (2016)²⁵ are shown in Figure 2 (red lines; overall $r^2 > 0.8$)”.*

L148-149: why are you reporting TOC and d13C for these 5 samples only? It would be very useful to report and discuss d13C for the entire data set.

Response: *We showed the ¹³C data profile in the attachment, and in the supplementary material of the new version.*

L150: given analytical uncertainty on d13C values, round all values to 1 decimal

Response: *We have made this change and now report corrected and kept the $\delta^{13}C$ data to 1 decimal point.*

[Redacted text block]

[Redacted text block]

[Redacted]

[Redacted]

[Redacted]

[Redacted]

[Redacted]

[Redacted]

[Redacted]

[Redacted]

[Redacted]

[Redacted]

[Redacted]

[Redacted]

[Redacted]

[Redacted]

[Redacted]

[Redacted]

[Redacted]

[REDACTED]

L230-234: these divisions are arbitrary and in my opinion not supported by the data presented in the paper the authors are referring to.

Response: *We modified the text as follows:* “The nature of OC (e.g., labile versus recalcitrant characteristics) can be indirectly justified based on thermal stability³⁴.”; *see line 234-235.*

L243: looking at figure 2, samples C and E do not appear to fall on the “linear trends” characteristic of the hemipelagic sedimentation, they have systematically higher 14C ages. How can they be considered “background” then?? This is a clear weakness in my opinion, as it stands it doesn’t look like the authors have actually analyzed the thermal reactivity (and associated 14C composition) of any truly “background” sediment.

Response: *We have already discussed reasons of systematically higher ¹⁴C ages among the five samples selected and processed for in-depth analysis. However, we point out that our interpretations are independent of this issue. Specifically, we reiterate the following points:*

- i) For the purpose of this paper, we focus on ¹⁴C offsets between thermal fractions in samples C and E in order to obtain a relative, rather than absolute, chronology.*
- ii) Systematically higher ¹⁴C ages occur within thermal fractions. The parallel lines between corresponding thermal fractions suggest that ¹⁴C aging (i.e., radioactive decay) in the thermal*

fraction is unaffected by methodological differences.

iii) Our study seeks to develop a chronologic approach through comparison with known events. The time offset of two known events match well with observed age offsets between corresponding thermal fractions from different sediment layers. This chronologic “ruler” thus holds promise in application to similar sequences, and serves as a highlight of our study.

OC fluxes: provide details regarding these estimates (specifically sedimentation rates). In general I am not a fan of calculating fluxes based on a 1D description of sediment accumulation.

Response: *We added the following in the new version (see line 410: “OC flux = TOC x density x sedimentation rate”. and in line 414 “Sedimentation rate was calculated based on the ratio of sediment depth spanning time intervals constrained by known events. Sedimentation rate for the 2011-earthquake turbidite sequence was calculated by the ratio of depth of sequence (~30 cm) to 1 year (time interval between 2011 and the collection year, ~30 cm/yr). Sedimentation rates for the other two earthquake turbidite sequences were estimated by multiplying ~30 cm/yr and each sequence depth (assuming that every earthquake sequence exhibits rapid sedimentation during one year).”*

Figure 3: The y axis needs to be normalized to account for aging (i.e. sample E was deposited more than 1000 years before sample A, obviously affecting the bulk ^{14}C difference between these samples). The x axis isn't optimal either. Peak height is never the best choice when looking at complex “chromatograms” because of non-linear effects. Areas would be much better. Figure 4: Here you should also normalize the x axis to account for the aging effect (see comment above).

Response: *The one of main objectives in showing Figure 3 is that it allows for examination of the relationship between the bulk ^{14}C content and major organic constituents of the sediments. In Figure 3, we find that the mixing ratio of the two major organic components (that are manifested as partially thermally resolved peaks) appears to exert strong control on bulk ^{14}C values. Here, bulk ^{14}C content (Fm) results incorporate ^{14}C decay (aging) since sediment deposition (e.g., the reviewer mentions sample E was deposited 1000 years before sample A). This influence of ^{14}C decay will be reflected (mirrored) in ^{14}C values of corresponding thermal fractions, and thus this does not influence our interpretation that “bulk ^{14}C content is controlled by the relative proportion of organic components with different ^{14}C ages” in line 198-199. With respect to the second point, while two peak height measurements indeed do not account for non-linear effects, we believe peak height ratios serve as simpler way to express this observation. We did attempt to resolve the two components through a Gaussian deconvolution approach, however we believe this approach is less optimal for expressing aging (^{14}C depletion during the lateral transport of bulk sediment and ^{14}C decay during the sedimentation). It would add layer of complexity to the description of our main findings and associated interpretation that we believe would not be of benefit to the reader. Similarly, in Figure 4, we prefer to use conventional ^{14}C age to report the ^{14}C age of each thermal fraction, and utilize the ^{14}C decay of background sediment to gain relative chronology.*

Reviewer #3: This is an interesting paper that presents a data set on the bulk radiocarbon (^{14}C) age of organic carbon in sediments within the Japan Trench along with the ^{14}C ages and stable carbon isotopic compositions of thermal decomposition fractions produced by Ramped Pyrolysis/Oxidation. The paper is novel both in its presentation of detailed information on the ages of earthquake-induced event deposits in such a deep marine (hadal) setting and especially because of the application of the cutting-edge RPO technique. The paper could potentially be of interest to a wide spectrum of researchers, including those with primary interests in paleoseismology, sedimentary geochronology, and marine carbon cycling.

Response: *We appreciate the reviewer's positive comments concerning our paper's novelty and of its general interest to the geoscience community.*

Notably, however, the abstract states that the findings will “shed new light on the nature, frequency, and magnitude of past tectonic events,” yet I see no obvious indication of any contributions to those solving questions. In my view, some of the conclusions of this paper are well supported, but others are in need of further explanation.

Response: *We deleted this sentence, and added the following in the abstract: “Our findings shed new light on links between tectonically-driven sedimentological processes and marine carbon cycling, with implications for carbon dynamics and burial in hadal environments”. Indeed, our bulk ^{14}C age measurements provide a means to identify the earthquake-triggered sediments through recognition of ^{14}C age anomalies in the profile. Information on the magnitude and frequency of sediment supply may also be gleaned from the thickness and number of such ^{14}C age anomalies.*

The authors show that the bulk ^{14}C ages of three event deposits, which record earthquakes that occurred in 869, 1454, and 2011 AD, are thousands of years older than their ages of deposition. Based on RPO analyses of 5 samples, they conclude that the bulk ^{14}C ages must reflect the mobilization during these earthquakes of terrestrially-derived organic carbon that was “pre-aged” in the water column or seabed. These are interesting findings and I have no quibble with these conclusions.

Response: *We appreciate these positive comments.*

What is less clear to me (and in fact is counter-intuitive), is how pre-aging and consequent loss of young labile OM would increase the heterogeneity of ^{14}C ages in earthquake deposits. A clearer explanation is required. Furthermore, in my view the idea that comparison of the ages of specific thermal fractions at various levels in stratigraphic sequences could be used to develop age models for deep sea sediments is interesting, but highly speculative given that it is based on results from only two samples. The fact that this doesn't appear to work at all with one of the thermal fractions (T3) is problematic and unexplained. Finally the suggestion that bomb carbon may influence the age of even the most refractory thermal fraction in the most recent (Tohoku) earthquake layer in the Japanese Trench is contrary to the prior conclusion that this fraction is highly aged and requires clarification as well.

Response: *With respect to the heterogeneity of ^{14}C ages, in order to highlight the enhancement heterogeneity (^{14}C age offset between T_1 and T_5 fractions of earthquake-triggered sediment intervals B and D), we have moved the sentence down to line 239-247 “Preferential degradation of young, labile OM may lead to an increase bulk ^{14}C ages. Figure 4 shows the ^{14}C age spectrum of thermal fractions among the five samples investigated. The ^{14}C age difference (^{14}C heterogeneity) between the most labile (lowest temperature, T_1) and the most refractory (highest temperature, T_5) fraction exceeds 4000 yr in sample B and D. In contrast, samples C and E (inferred “background” sedimentation, deposited around A.D. 915 and 869, respectively), exhibit smaller and more consistent age offsets (2714 \pm 118 yr and 2820 \pm 122 yr, respectively) (Fig. 4). Preferential degradation also enhances ^{14}C age heterogeneity (i.e., sample B and D).”*

While we are the first to admit that “the present dataset remains limited” (line 196), we are confident that our conclusions and approach utilizing ^{14}C ages of thermal fractions T_1 and/or T_5 of “background sedimentation” samples as “age ruler” to constrain chronologies of hadal zone sediments is robust. First, with respect to sample behavior, besides T_1 and T_5 fractions, T_2 fractions also exhibit “parallel” lines between sample C and E (see below Figure R2). Furthermore, if we draw connecting lines between T_1 and T_5 fractions in samples A and C, we obtain similar results. For example, the ^{14}C offset (1420 \pm 35 yr) between T_1 fractions in samples A and C is close to the corresponding known time offset between tectonic events (1096 yrs, from 2011 AD to 915 AD). Similarly, the ^{14}C offset (1056 \pm 191 yr) between T_5 fractions in the two samples is consistent with the known time offset between events. However, we note that the ^{14}C heterogeneity in sample A (^{14}C age offset between T_1 and T_5)

does not increase with respect to sample C. We attribute this different behavior to i) insufficient time for degradation of labile and younger OC due to the short interval between sampling and the tectonic event (only one year later after 2011 Tohoku), and/or ii) the possibility that “bomb” radiocarbon could influence the ^{14}C ages of thermal fractions in sample A, as mentioned in the main text. Thus, we find that “parallel” lines could also be drawn between sample A and C. Overall, although we discuss findings based on results from only two samples, we are confident that our approach has broader utility in constraining chronologies for hadal zone sediment sequences. In the future, we plan to test this dating approach on other sediment cores and further apply the RPO method at higher temporal resolution.

We also believe that there is no contradiction to the prior conclusion with respect to bomb spike effects on sample layers corresponding to the most recent (Tohoku) earthquake. The bomb spike could enrich ^{14}C content of organic matter, leading to younger apparent ages. The point here is that presence of ^{14}C aged (^{14}C -depleted) OC, for example in sample A, could be masked by contributions of bomb-derived radiocarbon. Even for fraction T_5 , we cannot exclude the possibility that some organic components contributing to this fraction are influenced by the bomb spike, although we agree that this fraction should be least influenced by bomb ^{14}C . At present, we lack a solid evident to exclude this possibility, and were therefore thought it prudent to consider this as a possibility. Future studies, potentially via compound specific radiocarbon analysis, may resolve this question.

With respect to the anomalous behavior of thermal fractions (T_3), it is important to note that these thermal fractions (391-476°C) fall into the temperature range in which they are affected by both of the two main components (the two distinct peaks, as indicated in Fig. 1). In order to be able to compare ^{14}C data among different sample fractions, we trapped CO_2 at fixed temperature intervals or “windows”, resulting in different mixing ratios of the two components in the T_3 fraction. Consequently, the T_3 fraction is sensitive to small variations in the proportions of the two fractions that result in variable ^{14}C ages between samples C to E. In contrast, T_1 and T_5 fractions are much less sensitive to variable mixing ratios and thus exhibit more uniform behavior. In line 311-314 of the new version of the manuscript, we have added the following: “the lack of observed consistency in ^{14}C ages of T_3 fractions is likely because this temperature fraction is sensitive to subtle variations in proportions of different organic components that contribute to this specific temperature interval (391-476°C).”

In line 315 of the previous version, we wrote: “Age models could be constructed provided these ^{14}C ages can be anchored to at least one sample of known calendar age (e.g., dated tephra layer)” which prompted the comment: “If we have the tephra we don't need to go to all this trouble.” In this study, the novelty of our chronological approach is that we (a) first identify which samples (“background” sedimentation) are appropriate for further chronological analysis through bulk OC ^{14}C measurement, and (b) use information derived from RPO to constrain ^{14}C age offsets (of T_1 and T_5 fractions) between sediment layers. This yields relatively ages based on one or more known events (absolute age). For example, the ^{14}C age of T_1 in sample A corresponds to the 2011AD Tohoku-oki earthquake is 1410 ± 15 yr BP, suggesting that all samples contain pre-aged OC that must be accounted for. While for absolute chronologies at least one sample of known calendar age constitutes a pre-requisite, the power of our approach is to document the timing and frequency of event deposits that pre-date historical records. Here, using our approach, we believe it will be feasible to extend chronologies prior to available historical records.

Figure R2. Chronology and radiocarbon characteristics of RPO thermal fractions of OM in sediment core GeoB 16431-1. Compared with Figure 4 in the main text, we linked the T₁ and T₅ fractions between sample A and C. In the main manuscript, we deleted the “parallel” lines, considering your suggestions.

The figures are good but would benefit from some improvements. The gray tones used to show background sediments and event layers in Figures 1 and 2 are too similar and are hard to see, and the blue bar is unexplained. Volcanic is misspelled in Figure 2.

Response: We add the caption concerning blue bar “Coccolith-bearing diatomaceous mud” in the Figure 1. We also deleted the lithology but kept the gray tones in Figure 2 of new version. Additionally, we also revised our manuscript based on the minor comments on typos and specific suggestions concerning the previous draft.

Reviewers' comments:

Reviewer #2 (Remarks to the Author):

I have carefully read the authors' response to the reviewer comments (mine and other reviewers') as well as the revised manuscript. While I commend the authors' detailed and careful response to the reviewer comments (providing additional graphs and all) I am left disappointed with the revision of the manuscript itself. In light of the authors' response, I stand by my initial assessment that the manuscript would be stronger if focused on C cycling (i.e. dropping the chronology component). First, there appears to be non-trivial mismatches between conventionally measured bulk composition and the bulk composition of the material the authors used for the RPO work. Stable isotopes are most affected (offsets in the 5 to 10‰ range!) but radiocarbon is also affected, albeit to a lesser extent. While this may not be an issue with respect to C cycle applications, it is obviously a problem with respect to deriving a precise and accurate chronology. Second, as it stands the value of the RPO isn't completely obvious besides identifying problematic samples. For "background" samples the ^{14}C age offset among for instance T1s is in the same ballpark as the offset between bulk compositions, what's the added value then? Further, comparing ^{14}C age offsets with calendar time differences is not straightforward. Additional specific comments/questions follow.

In general there is a need for more background information regarding sedimentation at the studied site. The authors refer repeatedly to a recently published paper (Ikehara et al 2016) but that doesn't help the reader who doesn't necessarily have the time to read it. In light of the bulk d^{13}C values the authors are now presenting and thanks to the clarification that samples C and E represent background sedimentation, I do wonder what the "background" sedimentation really is. It sure has more enriched stable isotope composition (which points towards marine organic carbon) but 1) the sedimentation rate for "background" periods is very high (e.g. 4.3 meters per kyr between 0.35 and 2.5 m!!) and, 2) the RPO thermogram of samples C and E don't really look very different from those of the other "turbiditic" samples (perhaps to the exception of sample D). To that end a cross plot of d^{13}C and F_m could be useful. In any case this point needs to be clarified in the manuscript.

Methodological issue: it looks like another difference between the 2 methods is the use of NaOH as a buffer after the fumigation for the samples prepped for RPO. This could potentially alter the organic carbon composition! Has this been tested?

Figure 3: I stand by my initial comment that the authors should normalize the F_m values to correct for aging. In their response they claim "it allows for examination of the relationship between the bulk ^{14}C content and major organic constituents of the sediments". In reality it's likely that aging (e.g. consider that sample E was deposited > 1000 years prior to sample A) alters the true relationship between ^{14}C content and composition (which is what the authors actually are after as it informs on the source and cycling on the organic carbon pool). For instance, samples E and B have very similar height ratio or peak 1 to peak 2 (close to a value of 1), consider that if one had been deposited say 10,000 years prior to the other they would have very different bulk ^{14}C ages and the correlation would completely fall apart!

Reviewer #3 (Remarks to the Author):

I have reviewed the revised manuscript and am satisfied that the authors have addressed the comments of the reviewers. I still think that Figure 1 could be made easier to look at if the gray tones used on the event layers had a greater contrast with the dark gray used for the background sediments. Other than that minor fix, I think it is in good shape. This is a very nice paper-- I look forward to seeing it published.

Response to Reviewers' comments

In order to further clarify our Responses in light blue, all responses that are not quotes from the main manuscript in the new version are written in *Italics*.

Reviewer #2(Remarks to the Author):

I have carefully read the authors' response to the reviewer comments (mine and other reviewers') as well as the revised manuscript. While I commend the authors' detailed and careful response to the reviewer comments (providing additional graphs and all) I am left disappointed with the revision of the manuscript itself. In light of the authors' response, I stand by my initial assessment that the manuscript would be stronger if focused on C cycling (i.e. dropping the chronology component).

Response: We appreciate these positive comments on our responses. While we also agree that the manuscript can be developed to a paper focusing on carbon cycling in the deep ocean, we here would like to reiterate the merits of this manuscript and other reviewers' comments. In the first review, it was commented that "This is a very nice study that tackles an important scientific question. I am impressed by the density of radiocarbon data (figure 2, middle panel is amazing!) and I find the thermal decomposition analysis novel and powerful." We present results from detailed radiocarbon-based investigation of the organic matter in a sediment core retrieved from the Japan Trench (> 7.5 km water depth). Indeed, our bulk ^{14}C age measurements provide a means to identify earthquake-triggered sediments through recognition of ^{14}C age anomalies in the profile. Information on the magnitude and frequency of sediment supply may also be gleaned from the thickness and number of such ^{14}C age anomalies. We develop an approach to establish the relative chronostratigraphic framework for hadal zone sedimentary records through a novel application of thermal decomposition analysis. Considering the spirit of Nature system journals, we wish to publish a paper of broadest interest to the geoscience community. For this reason, our desire is to retain the focus on the chronologic aspect of this study. We hope that you will agree that this is the optimal and most compelling framework for the manuscript.

Nevertheless, in order to clarify our arguments, we now further qualify the merits and limitations of the chronological approach (see line 426-453 in the new version).

Chronological considerations In the section of this paper addressing the development of Japan Trench sediment chronologies, we focus on ^{14}C offsets between thermal fractions (e.g., samples C and E) in order to derive a relative, rather than absolute chronology, and highlight its potential relevance to other hadal zone sedimentary sequences lying below the Calcite Compensation Depth that lack microfossils for conventional radiocarbon dating and isotope stratigraphy. Although the differences in isotopic compositions between bulk OC and RPO data are significant, they are roughly constant (small, F_m values. $\sim 0.06-0.08$, ± 0.02) (Supplementary Fig. S2A). As such, the isotopic discrepancy does not impact the chronological aspect of the paper and specifically the validity of chronologies developed based on RPO. Considering the mass contribution of target thermal fractions to each sample and the associated ^{14}C measurement errors (ave. $\pm \sim 100$ ^{14}C yr, based on ETH measurement), the offset (relative value) between ^{14}C ages of thermal fractions is relatively small ($\pm \sim 200$ ^{14}C yr), and does not undermine our statement in the manuscript. In the manuscript, we acknowledge this time difference (i.e., relative offset: 120 ± 45 yr, time offset: 46 yr). We emphasize that the novelty of our approach is to (i) first identify which samples ("background" sedimentation) are appropriate for further chronological analysis through bulk OC ^{14}C measurement, and (ii) use information derived from RPO to constrain ^{14}C age offsets (of T_1 fractions) between sediment layers. This yields relative ages that can be anchored based on one or more known events (calendar age). Furthermore,

while note that the presented ^{14}C ages are not calendar ages (documented event time), during the late Holocene (which constitutes the time interval of primary interest in this study), measured ^{14}C ages closely parallel calendar ages (ref. 40). Thus, accuracy of the relative chronology is dependent on measurement precision and offsets between calendar and ^{14}C ages, we consider this constitutes a unique approach for documenting the timing and frequency of event deposits pre-dating historical records in the hadal zone.”

First, there appears to be non-trivial mismatches between conventionally measured bulk composition and the bulk composition of the material the authors used for the RPO work. Stable isotopes are most affected (offsets in the 5 to 10‰ range!) but radiocarbon is also affected, albeit to a lesser extent. While this may not be an issue with respect to C cycle applications, it is obviously a problem with respect to deriving a precise and accurate chronology. Second, as it stands the value of the RPO isn’t completely obvious besides identifying problematic samples. For “background” samples the ^{14}C age offset among for instance T1s is in the same ballpark as the offset between bulk compositions, what’s the added value then? Further, comparing ^{14}C age offsets with calendar time differences is not straightforward.

Response: *We agree that the discrepancy in isotope values between the high-resolution bulk OC samples and the 5 samples selected for RPO was puzzling. We therefore decided to re-measure these 5 samples previously measured at WHOI using the identical protocol and apparatus that employed for the high resolution sample suite at ETH Zurich. As mentioned in the prior response, they were prepared in Ag capsules and acidified at ETH Zurich. The entire pre-treatment procedure is same as other high-resolution samples (background sample). The results show that the originally observed “mismatches” disappeared when we pre-treat these samples in a uniform way (see Figure).*

Figure. Left panel: previous ^{14}C age profile in which the five samples (different symbols) were pre-treated at NOSAMS, whereas “other samples” (black circles) were processed at ETH; Right panel: all the samples including the five samples (black symbols) after pre-treatment and analysis at ETH, with “Other samples” are shown as grey-filled circles.

Overall, we suspect that the reason for the overall discrepancy is due to the different sample

preparation/measurement methods applied. Specifically, the discrepancy may reflect a higher blank contribution in the NOSAMS method as the fumigation-based acid-treatment procedure was specifically implemented at NOSAMS for this study in order to maintain consistency between laboratories. The high-resolution samples were prepared in Ag capsules and acidified at ETH Zurich; whereas the five samples selected for detailed investigation were acidified in petri dishes prior to combustion to CO₂ and subsequent measurement at NOSAMS. An additional methodological difference is that the former the latter were first split on a vacuum line at NOSAMS and measured using ¹⁴C gas-measurement at ETH Zurich, while the latter were prepared and measured using a coupled elemental analyzer (EA) / IRMS / AMS online system at ETH Zurich (McIntyre et al., 2016).

Regarding the differences in bulk stable carbon composition, samples C and E used to establish the relative chronology exhibit a significant overall isotopic discrepancy. We believe this also derived from the two different pre-treatments methods, which were discussed in detail in our first response. This discrepancy in stable isotopic compositions also vanished with the re-run of the five samples (Supplementary Fig. S1). We are therefore confident in the new values that have been obtained, and conclude that the discrepancies arise from methodological differences between NOSAMS and ETH. Nevertheless, we emphasize that these differences do NOT impact the chronological aspects discussed in our paper as the relative chronologies are (i) only developed on the RPO data and (ii) only relative and not absolute ages (but are anchored at a petrologically and geochemically well-characterized tephra bed identified as the Towada-a tephra of the historical eruption of Towada volcano in AD 915). The ¹⁴C offsets for samples C and E are constant (Fig. 2, in the manuscript), and such constant offsets do not affect our relative chronology. Additionally, in the line 305-308, “We note that the ¹⁴C age difference (120 ±45 yr) between T₁ fractions in samples C and E approaches the time offset (46 yr) between the volcanic eruption in A.D. 915 and Jogan earthquake in A.D. 869 (Fig. 4)”. We thus note this difference/similarity between our T₁ ¹⁴C age offset and real age offset, but do not consider it appropriate to propose an absolute chronology for bulk sediments based on this approach without better understanding and constraints on local ¹⁴C reservoir effects and other uncertainties. Given the potential value of trench sediments as archives of past tectonic activity, extending beyond instrumental and historical records, we consider the chronological aspect of our study carries particular significance.

With respect to the comparison between calendar age and ¹⁴C age, this is common issue. In the manuscript, we have noted that direct use of conventional ¹⁴C ages of both bulk OC and T₁ fraction as an absolute dating tool is precluded. “¹⁴C age differences between specific thermal fractions may provide an approximation of age differences between sediment intervals, particularly for sediments deposited within the last ~3000 years, during which radiocarbon ages closely parallel calendar ages (Figure)” in line 302-305. We maintain that our novel approach to derive a relative chronology offers the potential to yield novel insights into the frequency of tectonic events, particularly over late Holocene.

Figure. ¹⁴C age (BP) vs. calendar age (BP). Modified from Reimer et al. (2013). The red line is IntCal 13.

Reference:

Reimer et al., (2013) *IntCal 13 and Marine 13 radiocarbon age calibration curves 0-50000 year cal BP. Radiocarbon*, 55, 1869-1887.

Additional specific comments/questions follow.

In general there is a need for more background information regarding sedimentation at the studied site. The authors refer repeatedly to a recently published paper (Ikehara et al 2016) but that doesn't help the reader who doesn't necessarily have the time to read it. In light of the bulk d13C values the authors are now presenting and thanks to the clarification that samples C and E represent background sedimentation, I do wonder what the "background" sedimentation really is. It sure has more enriched stable isotope composition (which points towards marine organic carbon) but 1) the sedimentation rate for "background" periods is very high (e.g. 4.3 meters per kyr between 0.35 and 2.5 m!!) and, 2) the RPO thermogram of samples C and E don't really look very different from those of the other "turbiditic" samples (perhaps to the exception of sample D). To that end a cross plot of d13C and Fm could be useful. In any case this point needs to be clarified in the manuscript.

Response: *According to the suggestion, we have added further information on the nature of sedimentation in the Japan Trench, with more detailed sediment description depicting the sedimentological differences between the "event deposits" and the "background sediment". Please see line 129-135 "The terminal basins along the trench floor accommodate the episodic deposition of fine-grained turbidites as graded fine-sand layers fining-upward into homogenous diatomaceous mud. The interseismic hemipelagic deposits that form with high sedimentation rates from 0.8 to > 3.0 m/kyr effectively cover earthquake-induced turbidites and volcanic ash layers and preserve the deposits as a geological record of large tectonic and volcanic events". We also revised the stratigraphic column in Figure 1 (following reviewer #3' suggestion) to graphically distinguish between the homogenous (=event) and bioturbated (background) sediments.*

In addition, in order to highlight the compositional difference between sample C and E (background sedimentation) compared to samples B and D, we plot $\delta^{13}\text{C}$ and Fm values (see Figure below), based on the reviewer's suggestion. We find that samples B and D (normal and inverted triangles in the Figure) exhibit marked scatter, particularly with respect to $\delta^{13}\text{C}$ values, whereas sample C and E generally cluster in one corner of the plot. While we agree that this figure does add a further layer of information, we do not believe it contributes to our main story. Thus, we prefer to include it as a supplementary Fig. S3.

Figure. Cross-plot between Fm values and $\delta^{13}\text{C}$ data of RPO fractions (T_1 - T_5) among the five samples.

Methodological issue: it looks like another difference between the 2 methods is the use of NaOH as a

buffer after the fumigation for the samples prepped for RPO. This could potentially alter the organic carbon composition! Has this been tested?

Response: *In order to avoid potential losses of soluble OC during the acid-rinsing process associated with carbonate removal, we used fumigated samples for RPO. We note that most published papers about RPO have utilized acid-rinsing for removal of inorganic carbon. We acknowledge that the fumigation pre-treatment method carries its own uncertainties. Thus, we have prepared a separate paper that provides a detailed discussion on the factors (e.g., carbonate concentration) that influence resulting isotopic data (Bao et al., L&O methods, in revision). Thus far, we have not tested the buffering role of NaOH, but the assumption is that the likelihood of C loss is minimized by avoiding rinsing procedures. Direct assessment of the influence of NaOH would be a useful future test of the overall methodology. However, we believe that maintaining uniform sample pre-processing facilitates consistent interpretation of the resulting dataset.*

We agree that open questions remain regarding the interpretation of our data. This is not surprising given the novelty of our approach, and regarding the currently limited knowledge of sedimentation processes in hadal environments such as the Japan Trench. We believe maintaining a broad perspective, including discussion concerning sediment chronologies, is warranted given the frontier-aspect of our study of sedimentation and carbon burial processes in the hadal zone. We consider that the remaining open questions, in our opinion, can be viewed in a positive sense, as it will motivate future research and follow-up investigations.

Reference:

Rui Bao, Ann P. McNichol, Jordon D. Hemingway, Mary C. Lardie Gaylord, Timothy I. Eglinton. The effect of different acid-treatments on the radiocarbon age spectrum of organic matter in sediments determined by Ramped PyrOx/Accelerator Mass Spectrometry. Submitted to Limnol. & Oceanogr., Methods.

Figure 3: I stand by my initial comment that the authors should normalize the Fm values to correct for aging. In their response they claim “it allows for examination of the relationship between the bulk ^{14}C content and major organic constituents of the sediments”. In reality it’s likely that aging (e.g. consider that sample E was deposited > 1000 years prior to sample A) alters the true relationship between ^{14}C content and composition (which is what the authors actually are after as it informs on the source and cycling on the organic carbon pool). For instance, samples E and B have very similar height ratio or peak 1 to peak 2 (close to a value of 1), consider that if one had been deposited say 10,000 years prior to the other they would have very different bulk ^{14}C ages and the correlation would completely fall apart!

Response: *We agree that “aging alters the true relationship between ^{14}C content and composition”. We yet believe that this is not in contradiction with our statement related to Figure 3. Both bulk and corresponding RPO fraction ^{14}C contents are affected by aging. During the burial/accumulation of sedimentary OM, a portion of the OC may undergo degradation, which would be reflected by a change in the shape of the thermogram (or peak ratio). Consequently, it is not straightforward to normalize the CO_2 curve based on ^{14}C age since it reflects a combination of degradation and post-depositional aging. We therefore believe that such a normalization would confound interpretation of the data. In addition, assuming for example that there is 10,000-year offset between any two samples (e.g., E and B), it would be reasonable to expect some diagenetically-driven OM alteration, resulting in changes in the thermogram (e.g., height ratio of peaks) as opposed to similar thermograms. We note, for example, that samples D and A exhibit distinct height ratios as well as contrasting Fm values, suggesting that changes of Fm values can reflect [diagenetically-induced] peak ratio variations. Acknowledging that while the present dataset remains limited, we believe Figure 3 highlights the relationship between ^{14}C contents and thermal shapes. Thus, we do not think that normalizing y-axis (Fm) is the most efficient way to*

convey our point in the manuscript. Moreover, since the age difference between samples E and B is only ~2 thousand years with modest associated changes in Fm values (0.15 ± 0.02 Fm), the proposed normalization would not result in a significant change in the linear relationship shown in Figure 3 or our interpretation. Nevertheless, taking into account the reviewer's comment, we now strengthen our argument by stating that "bulk ^{14}C content is largely controlled by the relative proportion of organic components" in line 203-204

Reviewer #3 (Remarks to the Author):

I have reviewed the revised manuscript and am satisfied that the authors have addressed the comments of the reviewers. I still think that Figure 1 could be made easier to look at if the gray tones used on the event layers had a greater contrast with the dark gray used for the background sediments. Other than that minor fix, I think it is in good shape. This is a very nice paper-- I look forward to seeing it published.

Response: *We appreciate these positive comments and the recommendation of this reviewer to publish our manuscript in a Nature Communications.*

In the new version, we have revised Figure 1 based on reviewer #1's #3's suggestion and now can also graphically better distinguish between the homogenous (=event) and bioturbated (background) deposits.

Reviewers' comments:

Reviewer #2 (Remarks to the Author):

The methodological issue has been adequately addressed at the bulk level (figure 2). Meanwhile, the fumigation procedure the authors used prior to RPO analysis clearly yields older ^{14}C ages and more depleted stable isotope compositions than conventional fumigation. I therefore wonder how much that can affect the RPO data. Figure S2 suggests differences of up to ca. 1000 ^{14}C years but it's unclear whether the offset is uniform across the temperature range or if any particular part of the thermogram (i.e. temperature fraction) could be specifically affected. Similarly, I wonder if the shape of the thermogram could be affected at all (which could impact the peak1/peak2 height ratio; figure 3).

Reviewer #2 (Remarks to the Author):

The methodological issue has been adequately addressed at the bulk level (figure 2). Meanwhile, the fumigation procedure the authors used prior to RPO analysis clearly yields older ^{14}C ages and more depleted stable isotope compositions than conventional fumigation. I therefore wonder how much that can affect the RPO data.

Response: We appreciate the reviewer's comments on this 2nd revised version of our manuscript, although it goes partly back to what we have already addressed in our previous responses, where we wrote: "*We suspect that the reason for the overall discrepancy is due to the different sample preparation/measurement methods applied*"; and "*the discrepancies arise from methodological differences between NOSAMS and ETH*". In our 3rd response here, we would like to (i) further discuss the challenges of applying across-lab experiments (which has been a long-standing issue that has garnered the attention of many geochemists) and (ii) underline that – even though this issue constitutes a true challenge – IT DOES NOT impact the scientific foundation of the paper (namely the chronological aspects), or the interpretations and conclusions:

Acid rinsing or acid fumigation, as common methods to remove inorganic carbon, both have disadvantages and advantages (*Komada et al., 2008; Brodie et al., 2011*). We agree that some organic matter will be solubilized during acid rinsing, as demonstrated by many studies (*Kennedy et al., 2005; Komada et al., 2008; Jaschinski et al., 2008; Brodie et al., 2011*), while fumigation approaches raise their own issues (*Komada et al., 2008*). Nevertheless, we do not think that it is essential to assess potential inter-lab methodological differences provided that a single approach is applied to generate a coherent data set, as is the case for our study. Taking the publication of Rosenheim et al. (2008) as an example, the issue of potential solubilization of organic matter associated with acid rinsing for removal of inorganic carbon did not affect the interpretation of the RPO results. We maintain that, provided the methodology employed is clearly described and all data were processed in a uniform fashion (as is the case in our study), the interpretations and conclusions are robust.

It would of course be worthwhile to further investigate the influence of different pre-treatment procedures on the organic carbon characteristics sediments, as well as to quantify inter-lab variations derived from application of different methodologies. However, this is a major undertaking which is beyond the scope of this study, and at present there is no consensus regarding the optimal pre-treatment method. Given that (1) we follow and fully describe previously published methodologies and apply them uniformly with respect to the specific suites of analytical procedures at NOSAMS and ETH, (2) we are NOT seeking to derive absolute values but to derive internally consistent data sets, and (3) our conclusions are based on the relative offsets of ^{14}C contents among thermal fractions, we consider our approach is robust and valid. Therefore, the observed slight differences in organic carbon characteristics stemming from the two labs/methodologies do NOT impact the chronological aspects that form the basis of this study.

Reference:

- Brodie, C. R., M. J. Leng, J. S. Casford, C. P. Kendrick, J. M. Lloyd, Z. Yongqiang, and M. I. Bird. 2011. Evidence for bias in C and N concentrations and $\delta^{13}\text{C}$ composition of terrestrial and aquatic organic materials due to pre-analysis acid preparation methods. *Chemical Geology*. 282: 67-83. doi:10.1016/j.chemgeo.2011.01.007.
- Kennedy, P., H. Kennedy, and S. Papadimitriou. 2005. The effect of acidification on the determination of organic carbon, total nitrogen and their stable isotopic composition in algae and marine sediment. *Rapid Communications in Mass Spectrometry*. 19: 1063-1068. doi: 10.1002/rcm.1889.
- Jaschinski, S., T. Hansen, and U. Sommer. 2008. Effects of acidification in multiple stable isotope analyses. *Limnology and Oceanography: Methods*. 6: 12-15. doi: 10.4319/lom.2008.6.12.
- Komada, T., M. R. Anderson, and C. L. Dorfmeier. 2008. Carbonate removal from coastal sediments for the determination of organic carbon and its isotopic signatures, $\delta^{13}\text{C}$ and $\Delta^{14}\text{C}$: comparison of fumigation and direct acidification by hydrochloric acid. *Limnology and Oceanography: Methods*. 6: 254-262. doi: 10.4319/lom.2008.6.254.

Figure S2 suggests differences of up to ca. 1000 ^{14}C years but it's unclear whether the offset is uniform across the temperature range or if any particular part of the thermogram (i.e. temperature fraction) could be specifically affected.

Response: We exclusively focus on ^{14}C age offset of the first thermal fractions between sample C and E in the main text. These thermal fractions for sample C and E in each case account for the same percentage (~33% mass%) of the bulk organic carbon. Given that the same proportions of thermal fractions of bulk OC are accounted in both samples and responding ^{14}C ages are corrected through the similar ^{14}C age difference of bulk measurements, the influence of ^{14}C age difference derived from different labs and methodologies in the specific or whole temperature range (thermal fraction) would be excluded through the ^{14}C age offset between the thermal fractions, therefore resulting in minimal impact on the data interpretation.

Due to different pre-treatment methodologies applied, it is not possible to determine whether RPO fraction prepared at NOSAMS exhibit systematically older ^{14}C ages or whether the bulk samples prepared and analyzed at ETH Zurich exhibit younger ages. As mentioned in the main text, considering the mass contribution of target thermal fractions to each sample and the associated ^{14}C measurement errors (ave. \pm ~100 ^{14}C yr), the offset (relative value) between ^{14}C ages of thermal fractions is relatively small (\pm ~200 ^{14}C yr), and does not undermine the arguments laid out in the manuscript.

[REDACTED]

[REDACTED]

[REDACTED]

[REDACTED]

[REDACTED]

[REDACTED]

[REDACTED]

[REDACTED]

[REDACTED]

REVIEWERS' COMMENTS:

Reviewer #3 (Remarks to the Author):

I am reviewing this paper for the third time and have enjoyed reading it again. In my opinion this paper is an important contribution to paleoseismology, to understanding carbon cycling in the deep sea, and to developing new approaches to geochronology in general. I have read through the authors replies to another reviewer's methodological concerns, and in my view their answers are adequate. At this point, I see no barrier to publication.

Reviewer #3 (Remarks to the Author):

I am reviewing this paper for the third time and have enjoyed reading it again. In my opinion this paper is an important contribution to paleoseismology, to understanding carbon cycling in the deep sea, and to developing new approaches to geochronology in general. I have read through the authors replies to another reviewer's methodological concerns, and in my view their answers are adequate. At this point, I see no barrier to publication.

Response: We appreciate the Reviewer #3's positive comments and all reviewers' efforts.